# Nutrient and Bioactive Composition of Five Gabonese Forest Fruits and Their Potential Contribution to Dietary Reference Intakes of Children Aged 1–3 Years and Women Aged 19–60 Years

**Robert Fungo** [1,2,*][iD]**, John H. Muyonga** [1]**, Judith Laure Ngondi** [3]**, Christian Mikolo-Yobo** [4][iD]**, Donald Midoko Iponga** [4]**, Alfred Ngoye** [4]**, Erasmus Nchuaji Tang** [3][iD] **and Julius Chupezi Tieguhong** [2,5]

1   School of Food Technology, Nutrition & Bio-Engineering, Makerere University, Kampala P.O. Box 7062, Uganda; hmuyonga@yahoo.com

2   Forest Genetic Resources Programme, Bioversity International, Via dei Tre Denari, 472/a 00057, Maccarese, Rome, Italy; chupezi@yahoo.co.uk

3   Department of Biochemistry, University of Yaoundé 1, Yaoundé, Cameroon; ngondijudithl@hotmail.com (J.L.N.); eras.tang@gmail.com (E.N.T.)

4   Institut de Recherche en Ecologie Tropicale (IRET), Centre National de la Recherche Scientifique et Technique (CENAREST), BP: 13 Libreville, Gabon; mick_jagg2001@yahoo.fr (C.M.-Y.); dmiponga@gmail.com (D.M.I.); alfredngoye@yahoo.fr (A.N.)

5   African Natural Resources Centre, African Development Bank, 01 BP 1387 Abidjan 01, Côte d'Ivoire

\*   Correspondence: rfungom@caes.mak.ac.ug

**Abstract:** Widely consumed forest fruits in Gabon were analysed for nutrient and bioactive compositions and their potential contributions to meeting the nutrient requirements of consumers. Edible pulps of *Panda oleosa* Pierre, *Gambeya lacourtiana* (De Wild.) Aubrév. & Pellegr. and *Poga oleosa* Pierre contained substantial amounts of bioactive compounds; flavonoids (13.5–22.8 mg/100 g), proanthocyanins (2.4–7.6 mg/100 g), polyphenols (49.6–77.3 mg/100 g) and vitamin C (6.7–97.7 mg/100 g). The highest content of β-carotene (76.6 μg/100 g) was registered in fruits of *Pseudospondias longifolia* Engl. The fruits of *P. oleosa* had the highest essential minerals Fe, Zn and Se. If a child aged 1 to 3 years consumed about 200 g or if a non-lactating and non-pregnant woman consumed 300 g of *Panda oleosa*, *Afrostyrax lepidophyllus* Mildbr., *G. lacourtiana*, *P. longifoli*a and *Poga oleosa*, they could obtain substantial DRI ranging between 20–100% for energy, vitamins C and E, iron, magnesium, iron and zinc. Forest fruits can considerably contribute towards the human nutrient requirements. Based on the results of this study, forest foods should be considered in formulating policies governing food and nutrition security in Gabon.

**Keywords:** wild fruits; phytochemicals; malnutrition and health

---

## 1. Introduction

In developing countries such as Gabon, forest foods provide income, essential nutrients and energy requirements to millions of urban and rural populations [1]. However, low intake of diversified foods in developing countries including Gabon, is among the top ten risk factors contributing to the health disorders related to inadequate intake of nutrients in Africa [2,3]. In Gabon, about 50% of the forests are allocated to logging concessionaires which are legislatively protected areas, thus limiting the surrounding populations' access to nutrient rich forest foods. The restriction is enforced by forest extension officers thus, affecting the food and nutrition security of the communities [4].

Rural poor populations are often among the most vulnerable ones because of the less diversified diets they consume, seasonality of foods they consume, intermittent drought, restricted participation of women in major economic activities of households, dysfunctional rural health services and lack of access to markets [3]. An estimated 75% of the population (including rural people) in Gabon predominantly consume processed cereals and meats [5]. According to the World Health Organization (WHO), a daily intake of over 400 g of fruits and vegetables per person has a potential to protect against diet-related non-communicable diseases and micronutrient deficiencies [6]. Micronutrient rich vegetables and fruits including the ones sourced from forests are among the least consumed foods by Gabonese, making their diets less diverse [5]. Non-diversified diets can result in negative consequences on an individual's wellbeing and development, because these diets may not meet essential nutrient requirements [7], exacerbating micronutrient deficiencies, obesity and non-communicable diseases [8]. Gabon is among the top ten countries in sub-Saharan Africa, with the highest prevalence of obesity and non-communicable diseases [5,8]. Available data reveals that overweight and obesity prevalence in Gabon has increased from 11.9% in 1995 to 20.4% in 2014 [5,8] with an average annual growth rate of 2.88%. This trend is more likely to increase if no alternative measures are taken. Furthermore, the high prevalence rates of Vitamin A deficiency (28%) and iron deficiency anaemia (72%) among Gabonese children are considered a public health problem because these rates are beyond the set range of 15% [9].

Promoting sustainable consumption of the readily available nutrient dense forest foods in Gabon could go a long way in addressing health disorders related to malnutrition. Sustainable consumption is when natural products such as forest foods are harvested for consumption, in order to bring a better quality of life to consumers, while minimizing the over exploitation of the forests so as not to jeopardize the needs of further generations [10]. Studies carried out in Cameroon [11,12] and Nigeria [13], emphasized the superiority of forest foods in contributing to essential nutrients and bioactive contents than the conventional and imported processed foods. For example, in Cameroon, forest foods were found to contribute to improved food and nutrition security of communities adjoining forests [11]. They have potential to address both macro and micronutrient deficiencies and non-communicable disease disorders related with inadequate intake of bioactive compounds [11,12]. In Gabon forest foods contribute about 82% of protein, requirements, 12% of energy, 36% of vitamin A and 20% of iron requirements [4]. In neighbouring Cameroon, forest fruits provide considerable amounts of vitamins C, A and E, selenium, calcium, iron and zinc to children aged between 1–3 years and women of reproductive age to promote health [12]. Some variations, due to genetic variability, or to differences in postharvest handling and stages of maturity, in nutrient content of fruits have been previously been documented [10]. On the other hand, forest foods also act as a safety net during times of severe food shortage, hence providing essential nutrients and food [14,15]. Furthermore, inadequate intake of flavonoids, polyphenols and proanthocyanins, vitamins C and E and essential minerals of zinc and selenium are risk factors for non-communicable diseases [4,5]. However, in Gabon, there is paucity of studies on nutrient composition and bioactive content of forest fruits. The primary aim of this study was to determine the nutrients and bioactive compounds content of five Gabonese forest plant foods. The five species were *Panda oleosa* Pierre, *Poga oleosa* Pierre, *Pseudospondias longifolia* Engl., *Gambeya lacourtiana* (De Wild.) Aubrév. & Pellegr. and *Afrostyrax lepidophyllus* Mildbr. The secondary objective of the study was to explore the potential nutritional contribution of these forest fruits to the recommended dietary intake (DRI) of both women and children.

## 2. Materials and Methods

### 2.1. Study Sites

The study samples were collected from villages surrounding two sites, which are owned by timber concessionaires. The first site located in the South-East was the "Compagnie Equatoriale des Bois (CEB)" owned by Precious Wood Inc. The second site located in the South West was the

Convention Provisoire Amènagement-Exploitation Transformation (CPAET) owned by Bayonne Inc (Figure 1). The CEB is located at latitude (φ): 00°83′36″ S; longitude (λ): 13°320′68″ E around Okondja town in the "Sebe Bricolo Departement" (Province of Upper Ogoouè). The forest concession covers an area of 615,000 hectares and employs about 1460 people. The population adjoining this timber concession is made up of about 14,000 people [16]. Their main source of income is chainsaw milling with agriculture poorly developed in this area [16,17]. Access to the forest concession by the surrounding populations' is legislatively limited, hence limiting the populations to collect the nutrient rich forest foods [4]. Fruit samples in this site were collected around the former Industrial License 2/90 that is located in the Eastern part of the concession. The second site (CPAET) located between the provinces of Nyanga and Ngounié covers an area of about 72,113 ha. In this forest concession, the total population is estimated at about 1600 people [17]. Due to limited employment opportunities around the forest concession, people primarily depend on the collection of forest foods around the previously logged forested areas that have been abandoned by concessionaire companies and related natural resources including forest fruits for household consumption.

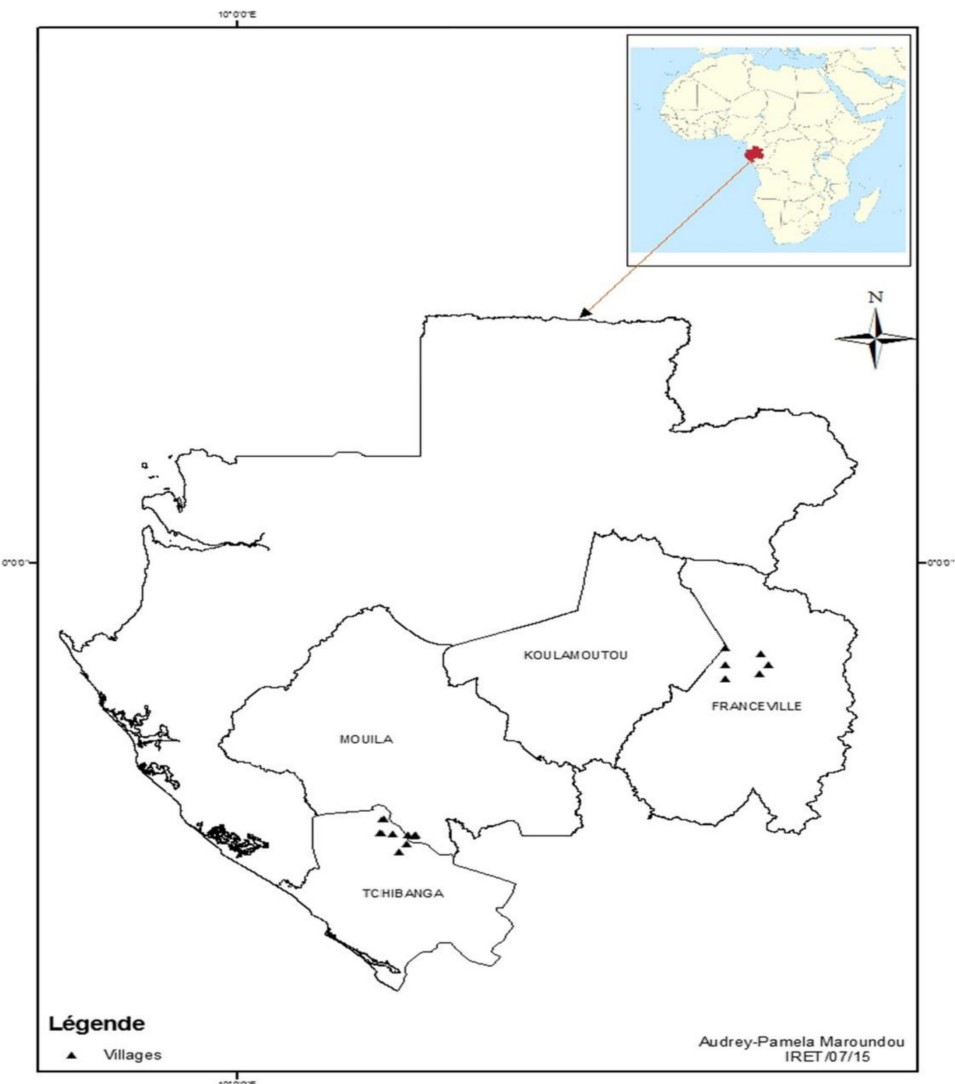

**Figure 1.** Study villages in two regions of the two timber concessions.

## 2.2. Forest Fruits and Nuts Sampling and Sample Preparation

Six readily available and widely consumed edible parts of forest fruits and nuts species including; *Panda oleosa*, *P. oleosa*, *P. longifolia*, *G. lacourtiana* and *A. lepidophyllus* were sampled from selected

villages in the two sites (Figure 2). All villages were located outside the concessions. The villages where fruit samples were collected were selected on the basis of their accessibility and proximity to the annually allocated timber logging areas. A multi-stage cluster sampling technique involving one stage of purposeful selection of districts and one stage of randomization of villages was employed. In the first stage the most accessible administrative districts within each site and fitting the village selection criterion listed above were purposefully selected. In the second stage, two villages in the East and four villages in the South East were randomly selected from the selected districts of site respectively. The villages in the present study were randomly chosen from the list of villages in each district that were accessible and located close to the annually allocated timber logging. From the South East, samples were collected from the villages of Mbouga and Otundou while in the South West, samples were collected from the villages of Okila, Tondondo, Dougassou and Nyangadougou. In Mbouga and Otundou, the fruits of *Panda oleosa* and *Poga oleosa* were referred to as "dibetou" and "afos" respectively, while in the Okila, Tondondo, Dougassou and Nyangadougou, the fruits of *P. longifolia*, *G. lacourtiana* and *A. lepidophyllus* were natively called "ofoss," "abami" and "ngali" respectively.

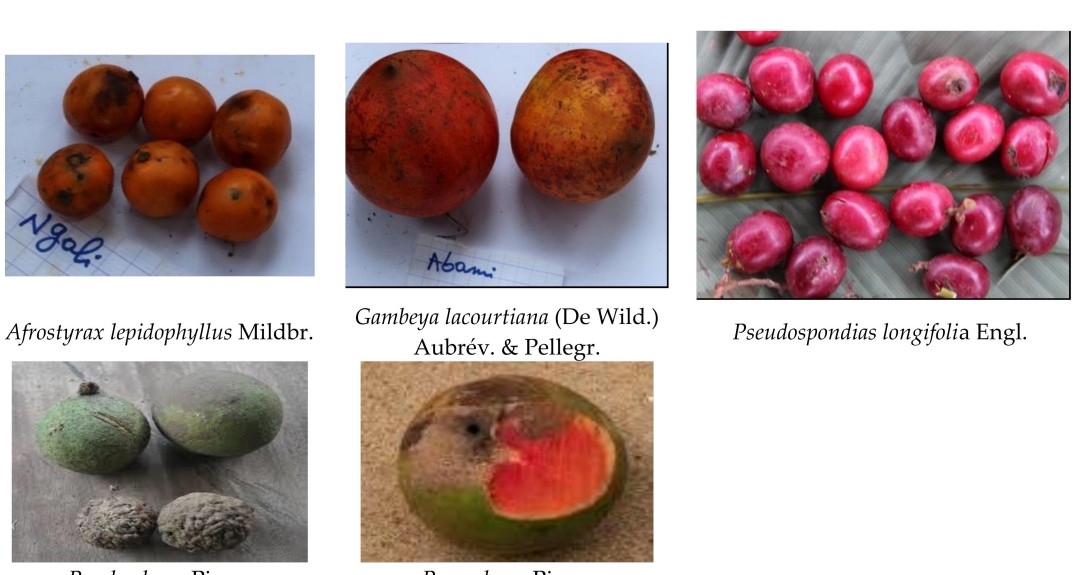

*Afrostyrax lepidophyllus* Mildbr.   *Gambeya lacourtiana* (De Wild.) Aubrév. & Pellegr.   *Pseudospondias longifoli*a Engl.

*Panda oleosa* Pierre   *Poga oleosa* Pierre

**Figure 2.** Fruits of A. lepidophyllus, G. lacourtiana, P. longifolia and nuts of P. oleosa and P. oleosa.

From each study village, five mature fresh fruits and nuts per species were sampled from different points of the surrounding forest and in accessible blocks that have been logged by concessionaire companies around each village and packaged in a perforated plastic container that was labelled and kept in an ice box container (Figure 3). The collected samples were then transported to the "Institut de Recherche en Ecologie Tropicale" (IRET), Libreville and stored in a refrigerator at 4 °C for one night. The next day, the samples were transported to Cameroon by air, for nutrient and bioactive compound analyses at the Nutrition and Biochemistry laboratory of the University of Yaoundé I. At the laboratory, the samples were washed with deionized water and conserved in a refrigerator at 4 °C. For each fruit species, two fruits out of the five per village were randomly selected for edible pulp for *A. lepidophyllus*, *G. lacourtiana* and *P. longifolia* and for nuts for *P. oleosa* and *P. oleosa.* The pulp and nuts from the species were mashed in a blender and then the extracts per species, were divided into three sub samples. The first sub sample was immediately sealed in clean polyethylene bags and conserved at −20 °C. The second fresh pulp was used for vitamin and bioactive compound analyses, while the third was analysed for moisture content. Dried samples were conserved for proximate analysis (excluding moisture content).

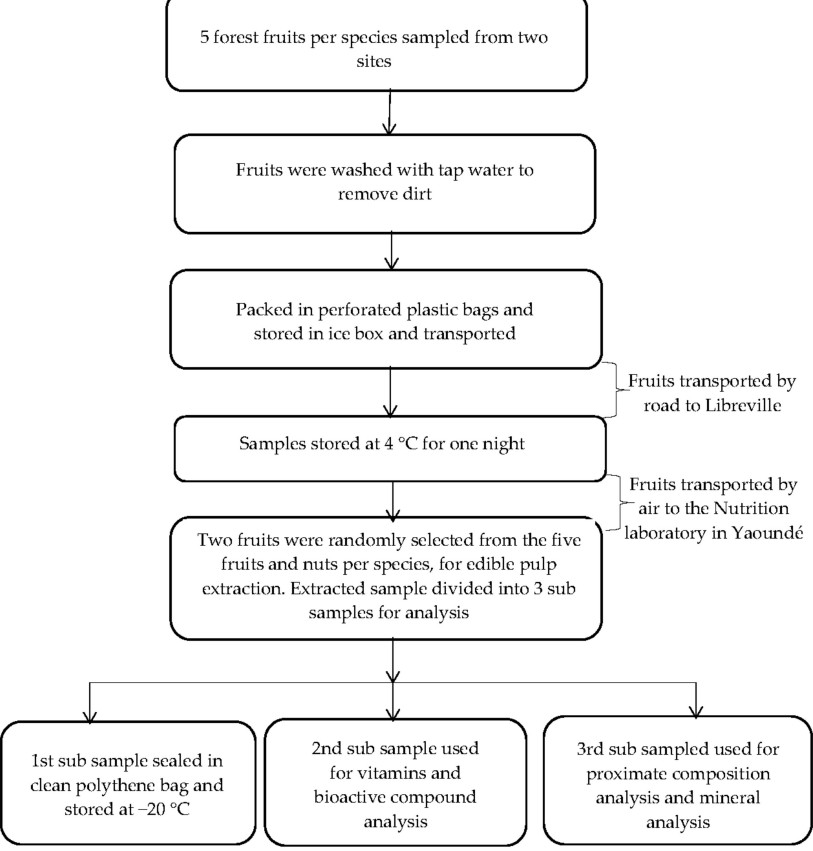

**Figure 3.** A schematic representation of the sampling of edible parts of fruits and nuts in the field and in the Laboratory.

### 2.3. Nutrients and Bioactive Compounds Analyses of the Extracted Fruits Pulps and Nuts

For nutrients and bioactive compounds analyses of the pulp and nuts extracts, the reagents, reference standards and solvents were purchased from Merck (Darmstadt, Germany). All the analyses were conducted in triplicate.

### 2.3.1. Proximate Analysis

The total moisture content was determined by measuring the weight loss of 5 g of fresh pulp and mashed nuts, using the vacuum oven method 934.01 [18]. The total fat content was determined using the ether extraction method 920.39 [18]. Crude fat was extracted from the fruit pulp and nut extract samples using 5 g of the extract in petroleum ether as a solvent and soxhlet extractor. After evaporation of the petroleum ether, the weight of the fat obtained gave the crude fat in the samples. The inorganic matter (total ash) was determined using method 942.05 [18]. The organic matter of the six samples was excluded by heating them at 550 °C overnight and the residue being the inorganic matter (ash). The total crude fibre content was determined using the Association of Official Analytical Chemists (AOAC) method 978.10 [18]. Defatted 5 g of extract pulp per species, were used to determine the fibre contents by acid digestion, filtration and base digestion. The resulting residues were eventually ignited at 550 °C in a Muffle furnace. Crude fibre content was expressed as a percentage lost on ashing, compared to the initial weight. While the total crude protein content was determined using the Kjeldahl 984.13 method [18]. 5 g of the fruit extract per species was digested with 5 mL of concentrated sulphuric acid and digested with a Kjeldahl digestor (Model Bauchi 430). A volume of 40 mL water was added and distilled using a Kjeldahl distillation Unit (Model unit B-316) containing 40% concentrated sodium hydroxide and Millipore water. Liberated ammonia was collected in 20 mL boric acid with bromocresol green and methyl red indicators and titrated against 0.04 N $H_2SO_4$. Crude protein was

calculated by multiplying by a factor of 6.25. The total carbohydrate content was calculated using the difference method of Greenfield & Southgate [19] as follows; 100% − (crude protein% + ash% + crude fat% + moisture%). In each edible portion, metabolizable carbohydrates were calculated following the method recommended by Greenfield & Southgate (2003): 100% − (crude protein% + ash% + crude fat% + moisture% + crude fibre%).

### 2.3.2. Minerals

For mineral content determination the fruit samples were digested using nitric/sulphuric acid (1:1 *v/v*) mixtures [18], prior to mineral content determination. The atomic absorption spectrophotometric method 975.03B [15], was used to determine Calcium (Ca), copper (Cu), magnesium (Mg) and zinc (Zn) contents. Iron (Fe) and selenium (Se) were determined using the atomic absorption spectrophotometric methods 999.11 [18] and 996.16 (G) [18] respectively. Potassium (K) and sodium (Na) were separately determined calorimetrically, using the flame emission photometric method 956.01 [18] while Phosphorous (P) content was determined using the gravimetric method 966.01 [18]. The atomic absorption spectrophotometer (AAS) and calorimeter were both calibrated against a series of standard solutions containing the elements under investigation. Mineral concentrations of each sample were quantified by comparison, against the standard curves. Standard curves were obtained after calibrations of each instrument was performed using a series of standards for each corresponding pure mineral of sodium, magnesium, iron, zinc, selenium, potassium and calcium and phosphorous and they were prepared from a 1000 ppm single stock solution made up with 2% nitric acid. The external calibrations were then run in the same analytical sequence as the samples [18].

### 2.3.3. Bioactive Compounds

Vitamin C, *β*-carotene, Vitamin E, Flavonoids, Polyphenols and Proanthocyanins

(20 g) of fresh edible part of each fruit and nut sample was used to extract the bioactive compounds. For total phenolic and total flavonoid contents analyses, the extraction procedure described by Fungo et al. [10] was used. The pulp and nuts were carefully removed from fruits and nuts respectively, grated and blended to fine pulp using a blender (Magic Line, Model MFP 000, Nu World Ind. (Pty) Ltd., Johannesburg, South Africa) with stainless steel blades. The fine pulp was stirred continuously with 50 mL (80% *v/v*) methanol for 24 h. The extract was filtered and the filtrate was centrifuged at 4000× *g* for 15 min using a centrifuge (Hitachi Model CR22N CE; Hitachi Koki Co., Ltd. Life Sciences Instruments, Tokyo, Japan). The supernatant was stored at 4 °C prior to use within two days. Total phenolic and total flavonoid contents in the methanolic extract were determined using the Folin-Ciocalteu's calorimetric method [20] and the aluminium chloride calorimetric method [21] respectively. The absorbance of phenolic compounds was measured at 765 nm while the absorbance of flavonoids, was measured at 415 nm using UV-VIS spectrophotometer (U-2001; Hitachi Instruments Inc., Tokyo, Japan). Quantification of total phenolic content was realized using the standard curve of gallic acid prepared in 80% methanol (*v/v*) and results were expressed in milligrams gallic acid equivalent (GAE) per gram fresh weight (fw) of fruits [22]. Quantification of total flavonoids was done on the basis of standard curve of quercetin prepared in 80% methanol and the results were expressed in milligram quercetin equivalent (QE) per gram fruit and nut fresh weight. The extraction of proanthocyanins was done using 70% (*v/v*) ethanol in water overnight at room temperature and the content in the extracts was determined spectrophotometrically by the pH differential method described by Giusti et al. [23]. The UV-visible spectrophotometer (U-2001; Hitachi Instruments Inc., Tokyo, Japan) was used to read absorbance at 530 and 700 nm. The total proanthocyanins content was calculated according to the standard curve of pure cyanidin-3-*O*-glucoside (Darmstadt, Germany) and expressed as cyanidin-3-*O*-glucoside (Cyn-3-*O*-G) mg/100 g fresh weight. β-carotene content was determined colorimetrically using AOAC method 970.64 [24] while vitamin E was determined

using Spectrophotometer (UV/VISIBLE) as previously described by Fungo et al. [10]. Extraction of β-carotene was done using xylene and separated using column chromatography, while vitamin E (tocopherol) was extracted using alcoholic sulphuric acid and absolute alcohol method 971.30 [24]. The β-carotene and vitamin E were determined by measuring absorbance at 470 nm and 270 nm respectively against blank samples. Standard curves were made with pure β-carotene standard and pure tocopherol and the results expressed as μg β-carotene equivalent and μg vitamin E equivalent per 100 g fresh fruit and nut powder. Vitamin C (ascorbic acid) was extracted using phosphotungstate reagent (PR) and quantified using spectrophotometric procedures at an absorbance of 515 nm against a blank [25]. Vitamin C content was calculated using the calibration curve of the standard reference of L-ascorbic acid.

*2.4. Data Analysis*

Statistical analyses were carried out using SPSS version 21 statistical software. The results were expressed as means ± standard deviations. A one-way ANOVAs was performed using the post-hoc multiple comparisons of means of nutrients, by Tukey's test at $p = 0.05$. The nutrient contribution of an average daily intakes of forest fruits to nutrient intake recommendations of a child aged 1–3 years and a non-lactating non-pregnant woman, were calculated and expressed as a percentage of the recommended dietary Intakes (DRI) [26,27]. The nutrient contributions were calculated based on total daily food intake estimations of forest populations in Cameroon reported by Fungo et al. [10,11] and the Congo Basin forest communities reported by Yamauchi et al. [28], where children aged 1–3 years consume about 200 g daily and non-lactating non-pregnant woman consume an average 300 g of food daily. Using these daily fresh fruits and nuts intake estimates per individual, the potential contributions of the forest foods on the daily nutrients requirements among children and adult women was calculated.

## 3. Results

*3.1. Proximate Composition*

Proximate composition of the three fruits and two nuts, varied significantly ($p \leq 0.05$) across the species (Table 1). *Poga oleosa* and *Panda oleosa* were found to have high fat and calorie contents. The fat content in the nuts of *P. oleosa* and *P. oleosa is* more than 4 folds that registered in *Pseudospondias longifolia*, *Afrostyrax lepidophyllus* and *Gambeya lacourtiana*. The fruits of *A.lepidophyllus* had exceptionally high proteins and dietary fibre content, while *G. lacourtiana* and *P. longifolia* were found to have substantial levels of digestible carbohydrates, with contents of about 60.0%.

*3.2. Minerals*

Generally, the fruits of *Panda oleosa* contained significantly ($p \leq 0.05$) high amounts of several essential minerals including; zinc, selenium and sodium (Table 2). Calcium, magnesium and iron were highest in the fruits of *A. lepidophyllus*. Generally, the fruits of *P. longifolia*, significantly ($p \leq 0.05$) contained lower amounts of minerals, than the rest of the fruits and nuts analysed in this study. However, phosphorous in the fruits of *P. longifolia* was considerably higher than the rest of fruits.

*3.3. Bioactive Compounds*

Vitamin C, β-carotene, Vitamin E, Flavonoids, Polyphenols and Proanthocyanins

The fruits of *G. lacourtiana* contained exceptionally significant ($p \leq 0.05$) contents of bioactive compounds including; vitamin C, flavonoids, polyphenols and vitamin E (Table 3). *Poga oleosa* was found to have considerably high contents of proanthocyanins and vitamin E. The highest β-carotene content in the present study registered in the fruits of *P. longifolia*, is about four folds higher than the content registered in the fruits of *P. oleosa* and *P. oleosa* and 12 folds higher the content registered in *A. lepidophyllus*.

**Table 1.** Mean and standard deviations of proximate composition 'wet basis'[1] of the edible parts of five forest fruits and nuts consumed in Gabon.

| Sample | Proximate Composition Mean Concentrations (%) [2*] | | | | | | | |
|---|---|---|---|---|---|---|---|---|
| | Water Content | Total Fat | Ash | Total Protein | Total Carbohydrates | Digestible Carbohydrates | Total Energy (Kcal) [3] | Total Dietary Fibre |
| *Panda oleosa* Pierre | 13.4 ± 2.5 [a] | 53.1 ± 8.3 [a] | 6.8 ± 0.4 [b] | 4.2 ± 1.1 [a] | 35.9 ± 9.2 [a] | 30.3 ± 9.2 [a] | 615.9 [c] | 5.6 ± 0.7 [a] |
| *Poga oleosa* Pierre | 15.3 ± 0.7 [a] | 68.5 ± 6.4 [a] | 2.1 ± 0.5 [a] | 2.1 ± 0.8 [a] | 27.2 ± 5.9 [a] | 21.1 ± 5.9 [a] | 709.3 [c] | 6.1 ± 0.1 [a] |
| *Pseudospondias longifolia* Engl. | 29.0 ± 2.0 [b] | 11.1 ± 0.3 [b] | 6.0 ± 1.3 [b] | 2.7 ± 0.4 [a] | 85.8 ± 0.5 [b] | 66.4 ± 0.7 [c] | 376.3 [b] | 19.4 ± 0.5 [b] |
| *Gambeya lacourtiana* (De Wild.) Aubrév. & Pellegr | 54.9 ± 0.1 [c] | 13.6 ± 3.6 [b] | 2.9 ± 0.7 [a] | 4.3 ± 0.4 [a] | 79.2 ± 3.9 [b] | 59.6 ± 8.0 [c] | 378.0 [b] | 19.6 ± 4.0 [b] |
| *Afrostyrax lepidophyllus* Mildbr. | 4.6 ± 0.7 [d] | 17.4 ± 1.5 [b] | 7.9 ± 0.3 [b] | 11.9 ± 0.8 [b] | 62.8 ± 2.2 [b] | 4.6 ± 1.7 [b] | 222.6 [a] | 58.1 ± 0.8 [c] |

[1]: Composition of edible parts wet basis. [2]: Each value is the mean and standard deviation (SD) of 3 sample lots per species analysed individually. [3]: Metabolizable energy (Kcal) provided by substrates in diets = Digestible carbohydrates. × 4+ Total fat × 9 + Total proteins × 4 [10]. *: Within each column, means with different superscripts letters are significantly different as tested by Tukey's ($p < 0.05$).

**Table 2.** Mean concentrations (mg/100 g) edible parts wet basis[1] and standard deviations of common forest fruits and nuts consumed in Gabon.

| Sample [1] | Mean Concentrations (mg/100 g) [2*] | | | | | | | |
|---|---|---|---|---|---|---|---|---|
| | Na | K | Ca | Mg | P | Fe | Zn | Se [3] |
| *Panda oleosa* | 25.4 ± 1.5 [a] | 0.004 ± 0.01 [a] | 0.003 ± 0.0006 [a] | 10.5 ± 1.3 [a] | 0.005 ± 0.0008 [a] | 17.1 ± 1.3 [a] | 10.2 ± 0.03 [a] | 0.7 ± 0.1 [a] |
| *Poga oleosa* | 20.4 ± 0.01 [a] | 0.002 ± 0.0003 [a] | 8.1 ± 0.01 [b] | 10.9 ± 0.01 [a] | 0.002 ± 0.0004 [a] | 20.6 ± 0.6 [a] | 7.5 ± 0.1 [a] | 0.1 ± 0.01 [b] |
| *Pseudospondias longifolia* | 18.7 ± 0.02 [a] | 0.006 ± 0.003 [b] | 41.6 ± 0.9 [c] | 21.3 ± 0.3 [b] | 35.3 ± 0.07 [b] | 4.4 ± 0.01 [b] | 0.2 ± 0.01 [b] | 0.03 ± 0.002 [c] |
| *Gambeya lacourtiana* | 37.9 ± 1.04 [b] | 0.007 ± 0.0007 [b] | 0.004 ± 0.0004 [a] | 0.002 ± 0.4 [c] | 0.002 ± 0.003 [a] | 2.4 ± 0.1 [b] | 6.6 ± 0.3 [a] | 0.2 ± 0.02 [b] |
| *Afrostyrax lepidophyllus* | 8.7 ± 0.5 [c] | 0.002 ± 0.0003 [a] | 71.5 ± 1.9 [d] | 88.5 ± 1.9 [d] | 0.6 ± 0.05 [c] | 23.5 ± 1.7 [a] | 0.9 ± 0.04 [c] | 0.02 ± 0.002 [c] |

[1]: Composition of edible parts wet basis. [2]: Each value is the mean and standard deviation (SD) of 3 sample lots per species analysed individually. [3]: μg/100 g. *: Within each column, means with different superscripts letters are significantly different as tested by Tukey's ($p < 0.05$).

**Table 3.** Mean concentrations (mg/100 g) edible parts wet basis[1] and standard deviations of bioactive compounds of common forest fruits and nuts consumed in Gabon.

| Sample [1] | Mean Concentrations (mg/100 g) [2*] | | | | | |
|---|---|---|---|---|---|---|
| | Vitamin C | β-carotene [3] | Vitamin E [3] | Flavonoids [4] | Polyphenols [5] | Proanthocyanins |
| *Panda oleosa* | 6.7 ± 0.1 [a] | 17.2 ± 0.5 [a] | 23.2 ± 1.3 [a] | 13.5 ± 0.5 [a] | 49.6 ± 2.4 [a] | 7.6 ± 0.2 [a] |
| *Poga oleosa* | 4.6 ± 0.04 [a] | 17.1 ± 0.1 [a] | 21.4 ± 0.1 [a] | | | |
| *Pseudospondias longifolia* | 36.3 ± 0.01 [b] | 76.6 ± 4.3 [b] | na. | 6.5 ± 0.9 [b] | 42.6 ± 3.3 [a] | 1.1 ± 0.2 [b] |
| *Gambeya lacourtiana* | 97.7 ± 1.6 [c] | 0.003 ± 8.4 [c] | 19.4 ± 0.9 [a] | 22.8 ± 0.5 [c] | 77.3 ± 3.1 [b] | 2.4 ± 0.1 [b] |
| *Afrostyrax lepidophyllus* | 2.05 ± 0.1 [a] | 5.7 ± 0.1 [d] | 0.5 ± 0.002 [b] | 3.8 ± 0.1 [d] | 47.6 ± 0.7 [a] | 1.8 ± 0.1 [b] |

[1]: Composition of edible parts wet basis. [2]: Each value is the mean and standard deviation (SD) of 3 sample lots per species analysed individually. [3]: µg/100 g. [4]: Milligrams gallic acid equivalent (GAE) per gram fresh weight. [5]: Milligram quercetin equivalent (QE) per gram fruit weight. *: Within each column, means with different superscripts letters are significantly different as tested by Tukey's ($p < 0.05$).

## 4. Discussion

### 4.1. Proximate Composition

The fat, digestible carbohydrates and total energy contents in the nuts of *Panda oleosa* and *Poga oleosa* were significantly higher than the content registered in the three fruits of *Pseudospondias longifolia*, *Gambeya lacourtiana* and *Afrostyrax lepidophyllus*. Previous findings in *P. oleosa* from the Umudike rain forest and Oban National Park in Nigeria [29] revealed a fat content of 68.2% and carbohydrate content of 25.38%, falling in the range of the nuts of *Panda oleosa* and *Poga oleosa*, in the present study. Then again, when compared to popular and widely consumed oil producing oil seeds such as soy beans (*Glycine max*) in Gabon and the Congo Basin forest region, it was found that the lipid content of both *P. oleosa* and *P. oleosa* nuts to be about 3 folds higher than the fat content reported in *G. max* (23.5%) [30]. Based on their high nutrient content, the species of *P. oleosa* and *P. oleosa* can be considered good sources of fat and dietary energy and could play a role in production of high energy foods such as complementary foods for weaning children and foods to address severe acute malnutrition in emergency situations among forest communities of sub Saharan Africa. The highest protein and fibre contents registered in the present study in *A. lepidophyllus* were higher than the contents of protein (7.7%) and fibre (4.4%) reported in *A. lepidophyllus* grown in Southern Nigeria [31]. The difference between the contents of protein and fibre in the fruits of *A. lepidophyllus* of the present study in Gabon and that of Ene-Obong et al. [28], may possibly be due to differences in growth conditions, genetic variation, or due to differences in post-harvest handling, processing, storage conditions and stage of maturity [32]. The forest fruits of *P. longifolia* and *G. lacourtiana* were found to contain remarkably high amounts of carbohydrates. The carbohydrate content in these two forest fruits were higher than some commonly consumed fruits in Gabon and the other Congo basin countries, such as bush mangoes (*Irvingia gabonensis*) (13.5%) and bush butter fruits (*Dacryodes edulis*) (8.7%) as reported by Jamnadass et al. [33] and *Ricinodendron heudelotii* (34.1%) commonly referred to as "njangsang" in Cameroon and Nigeria as reported by [31]. However the protein and fat content of the fruits of *P. longifolia* and *G. lacourtiana* are considerably lower than the contents reported in *I. gabonensis*, *D. edulis* and *R. heudelotii*.

### 4.2. Minerals

Overall, the mineral contents in the nuts of the forest fruits of *Poga Oleosa* in the present study were in the range of the mineral contents reported in *Poga Oleosa* [29] and commonly consumed forest fruits; *I. gabonensis* and *D. eduli* [33]. The essential minerals of iron, zinc, calcium, magnesium and sodium in the fruits of *Poga Oleosa* of the present study, were in the range of the previous findings of the same species. A range of 22.1–22.2 mg/100 g for iron, a range of 6.68–6.72 mg/100 g for zinc, a range of 8.9–9.2 mg/100 g for calcium and a range of 10.0–10.12 mg/100 g for magnesium has been reported in *Poga Oleosa* growing in Umudike rain forest and Oban National Park of Nigeria [29]. Whereas the mineral contents in the fruits of *P. longifolia* and *G. lacourtiana* fall in the range of commonly consumed forest fruits such as the African pearwood (*Bailonella toxisperma*), a number of specific minerals were observed to be substantially lower than the contents reported in the commonly consumed fruits of the African oil bean (*Pentaclethra macrophylla)* and *Trichoscypha abut* [10]. For example the calcium, iron, zinc and selenium contents registered in the fruits of *P. longifolia* are comparable with calcium (37.5 mg/100 g), iron (3.3 mg/100 g), zinc (0.2 mg/100 g) and selenium (0.1 µg/100 g) contents reported in *B. toxisperma* [10]. However the potassium and phosphorous, contents in the five forest fruits in the present study, are lower than contents reported in *B. toxisperma* and *P. macrophylla* [10].

The iron, zinc and selenium contents registered in the fruits of *A. lepidophyllus* of the present study were comparable to contents registered previously in the same species harvested in Southern Nigeria [31]. Ene-Obong et al. [31] documented iron content of 24.3 mg/100 g, zinc content of 5.7 mg/100 g and selenium content of 0.01 µg/100 g in Nigerian *A. lepidophyllus* fruits. Nonetheless, the contents of calcium, potassium, sodium, phosphorous and magnesium previously reported in the

Nigerian fruits of *A. lepidophyllus* by Ene-Obong et al. [31] are remarkably higher than the contents of the same species reported in the present study. The difference between the contents from the two studies may be attributable to genetic variation, or probably due to differences in post- harvest handling and stage of maturity [32].

*4.3. Bioactive Compounds*

Overall the fruits of *G. lacourtiana* and *Panda oleosa* contained considerably higher contents of flavonoids, proanthocyanins, polyphenols, vitamins C and E, than the values reported for some forest foods and commonly consumed plant foods in Gabon.

Vitamin C

The highest vitamin C content of the fruits of *G. lacourtiana* is more than 13 times higher than the content reported in forest nut of *R. heudolitii* (7.5 mg/100 g), more than 11 times higher than the content reported in the fruits of *Tamarindus indica* (9.0 mg/100 g), more than three times the content in fruits of *D. eduli* (32.1 mg/100 g) and about two times higher than the content in the forest fruits of *I. gabonensis* (66.4 mg/100 g) and *Sclerocarya birrea* Hochst (68.0 mg/100 g) [33–35]. *G. lacourtiana* vitamin C content recorded in this study is about ten folds higher than the content reported in dessert bananas (*Musa* acuminate) (9.7 mg/100 g) and about two folds higher than that of papaya (51.2 mg/100 g) [36].

Flavonoids

The highest flavonoid content in the fruits of *G. lacourtiana* was comparable or considerably higher than the flavonoid content in some popular wild forest foods in West and Central Africa [37,38]. Flavonoid content in the fruits of *G. lacourtiana* of the present study falls in the range of previously reported contents of several forest fruits including; *Vitellaria paradoxa* (20.7 mg/100 g), *Adansonia digitata* (31.7 mg/100 g), *Dialium guineense* (19.45 mg/100 g) and *Diospyros mespiliformis* (22.4 mg/100 g). It is also about 14, 13, 10 and 2 times the values reported in the forest fruits of *Saba Senegalensis* (1.7 mg/100 g), *T. indica* (2.2 mg/100 g) and *Gardenia erubescens* (11.7 mg/100 g) respectively.

Phenolics

The fruits of *G. lacourtiana* had the highest phenolic contents. However, the five sampled fruits had low phenolic contents, as compared to the widely consumed forest fruits in Gabon and the rest of the Congo basin forest region [10,37,39]. For instance, previous studies have registered remarkably high phenolic contents of 300.0 mg/100 g in bush butter fruits of *D. edulis*, 364.0 mg/100 g in *Ricinodendron heudelotii*, 671.8 mg/100 g in the African oil fruits of *P. macrophylla*, 686.7 mg/100 g in the African pear fruits of *B. toxisperma*, 947.0 mg/100 g in *T. abut*, 957.3 mg/100 g in *T. indica*, 1347 mg/100 g in *Aframomum citratum* and 3518.3 mg/100 g in *A. digitata* [10,37–39]. The phenolic content of the fruits of *A. lepidophyllus* growing in Gabon in the present study, is 5 times less the content (212.0 mg/100 g) reported in *A. lepidophyllus* growing in natural habitats of Yaoundé natural forests of Cameroon [39], while on the other hand, it is 4 times higher than the content (12.0 mg/100 g) reported in similar species grown in Southern Nigeria [31]. The variation in the contents from the three studies can be attributed to differences in post- harvest handling of samples, stage of maturity or genetic variation [32]. Additionally, the phenolic contents in the forest fruits, in the present study, are comparable to the contents previously reported in common tropical and temperate fruits including; mangoes (*Mangifera Indica)* (72.2 mg/100 g), apples (*Malus pumila*) (45.8 mg/100 g) papaya spp. (75.4 mg/100 g), pineapples (*Ananas comosus*) (66.3 mg/100 g[1]) and Cavendish bananas (44.5 mg/100 g) [36,40,41]. The phenolic contents of forest fruits in this study reveal that these fruits can be alternative source of bioactive phenolic phytochemicals that can provide essential nutraceutical benefits similar to common tropical fruits.

Vitamin E

The fruits of *Panda oleosa* contained the highest vitamin E and proathocyanins contents. The vitamin E content registered in the fruits of *Panda oleosa* was considerably higher than the content reported in some of the forest fruits including; *T. abut* (0.02 µg/100 g) and *B. toxisperma* (9.3 µg/100 g) from Cameroonian forests [10] and *Parkia biglobosa* (18.1 µg/100 g) from Nigeria [42]. Furthermore, the vitamin E content in *Panda oleosa*, *Poga oleosa* and *G. lacourtiana*, are 20 times higher than, the content previously reported in imported fruits including; mangoes (*M. indica*) (1.1 µg/100 g) [43]. The vitamin E content in the three fruits was more than 20 times the contents in the citrus fruits of lemon (*Citrus limon* (L.) Osbeck), oranges (*Citrus reticulate*), tangerine (*Citrus Reticulate*) and *Vitis vinifera* (grapes), which have negligible contents [44].

Proanthocyanins

The results of this study revealed that the five forest fruits were not good sources of proanthocyanins and β-carotene. For example the highest registered proanthocyanins content (7.6 µg/100 g) in the fruits of *Panda oleosa* is about 9 folds less, the content previously reported in Cameroonian forest fruits of *P. macrophylla* (65.0 µg/100 g) and *T. abut* (61.2 µg/100 g) and four folds less the content in *B. toxisperma* (28.0 µg/100 g) [10].

β-carotenoid

The highest β-carotenoid content in the present study registered in the fruits of *P. longifolia*, is remarkably lower than the values reported in commonly consumed β-carotene rich foods such as papaya (232.3 µg/100 g), dessert bananas (96.9 µg/100 g) and East African highland cooking bananas (*Musa* spp.) (337.0 µg/100 g) [36,45]. The low levels of proanthocyanins and β-carotene in the present study may be attributed to the prolonged hours of transporting samples from the remote sites in Gabon to the nutrition laboratory at Yaoundé I University in Cameroon, for analyses. As a result of the unsaturated chemical structures of proanthocyanins and β-carotene compounds, there exposure to light and oxygen, may lead to isomerization and oxidation, thus reducing their contents in food samples [22].

Despite the variation in the concentration of the bioactive compounds in the fruits and nuts of the present study, these compounds have potential to promote antioxidant activity, a precursor to prevent diseases mediated by oxidative stress [46]. Furthermore, pharmacological studies reveal that flavonoids, phenols and proanthocyanins, at low levels, have the potential to substantially reduce diarrhoea and diabetes among humans [47]. The leaves and seeds of *G. lacourtiana* are used in the treatment of anaemia and stomach ache among the rural communities of Gabon while the *A. lepidophyllus* is used in the treatment of postpartum infections among women [48]. Also, the oil producing seeds of *B. toxisperma* are used by the Baka pygmies in both Cameroon and Gabon, to treat rheumatism and child birth shocks [10,49].

*4.4. Impact of the Nutritional Value of Forest Fruits on Dietary Essential Nutrient Intake*

The amount of foods (such as forest fruits) eaten traditionally by forest populations in Gabon, is estimated at around 200 g and 300 g daily respectively for children aged 1–3 years and non-lactating and non-pregnant women aged between 19 and 60 years [10,28]. Forest fruits in the Congo basin countries including Gabon and Cameroon are often eaten daily as snacks by children or by adults in between main meals or while performing household income activities such as farming or hunting [11,50]. Accordingly, using these estimations for the amounts of fruits that may be consumed, we estimated the potential contributions forest foods could make to meet the DRI of essential nutrients [51,52].

The calculated potential contribution of the five forest fruits in the present study in reference to intakes of the energy and key essential nutrients such as magnesium, iron, zinc, vitamins C and E,

revealed that these foods can provide considerable amounts of nutrients to both women and children (Table 4). For example, if a non-lactating and non-pregnant woman and a child aged 1 to 3 years ate 300 g and 200 g respectively daily, they would both meet their 100% DRI, for magnesium (1000 mg/day for women and 500 mg for children), iron (58.8 mg/day for women and 11.6 mg/day for children) and zinc (12 mg/day for women and 4.5 mg/day for children), from consumption of either *P. oleosa*, *P. oleosa*, *G. lacourtiana* and *A. lepidophyllus*. Also, 100% DRI for vitamin E requirement of 0.4 mg/day for children and 19 mg/day for women would be met by the fruits of *P. oleosa*, *P. oleosa* and *G. lacourtiana*. About 30% to 100% of the DRI for energy (1236.5 Kcal for children and 3000 Kcal for very active African women) and vitamin C (45 mg/day for women and 30 mg/day for children) would also be met by consumption of 300 g and 200 g of *Poga oleosa*, *Panda oleosa*, *P. longifolia* and *G. lacourtiana* by women and children, respectively. However, studies on bioavailability of the nutrients in these foods have not yet been carried out. Thus, further studies are needed to confirm the contribution of these foods to meeting the recommended intake for bioactive compounds.

**Table 4.** Estimated nutrient contribution of the average daily fruit intake [1] to the dietary recommended intakes (DRI) for children aged 1–3 years and women aged 19–60 years.

| Nutrients | Unit | *Panda oleosa* Pierre | *Poga oleosa* | *Pseudospondias longifolia* | *Gambeya lacourtiana* | *Afrostyrax lepidophyllus* |
|---|---|---|---|---|---|---|
| Energy (Kcal) [2] | % DRI 1–3 year | 99.6 | 114.7 | 60.8 | 61.1 | 36.0 |
| | % DRI 19–60 year | 61.6 | 70.9 | 37.6 | 37.8 | 22.3 |
| Na | % DRI 1–3 year | 12.7 | 10.2 | 9.4 | 18.9 | 4.4 |
| | % DRI 19–60 year | 15.2 | 12.2 | 11.2 | 22.7 | 5.2 |
| K | % DRI 1–3 year | 0.0005 | 0.0003 | 0.0008 | 0.0009 | 0.0003 |
| | % DRI 19–60 year | 0.0006 | 0.0003 | 0.0009 | 0.0011 | 0.0003 |
| Ca | % DRI 1–3 year | 0.0 | 3.24 | 16.6 | 0.00 | 28.6 |
| | % DRI 19–60 year | 0.0 | 2.4 | 12.5 | 0.00 | 21.5 |
| Mg | % DRI 1–3 year | 35.0 | 36.3 | 71.0 | 0.01 | 295.0 |
| | % DRI 19–60 year | 14.3 | 14.9 | 29.1 | 0.00 | 120.7 |
| Fe | % DRI 1–3 year | 294.8 | 355.2 | 75.9 | 41.4 | 405.2 |
| | % DRI 19–60 year | 87.2 | 105.1 | 22.5 | 12.2 | 119.9 |
| Zn | % DRI 1–3 year | 453.3 | 333.3 | 8.9 | 293.3 | 40.0 |
| | % DRI 19–60 year | 255.0 | 187.5 | 5.0 | 165.0 | 22.5 |
| Se | % DRI 1–3 year | 10.3 | 1.5 | 0.4 | 2.9 | 0.3 |
| | % DRI 19–60 year | 11.0 | 1.6 | 0.4 | 3.0 | 0.3 |
| Vitamin C | % DRI 1–3 year | 44.7 | 30.7 | 212.0 | 651.3 | 14.1 |
| | % DRI 19–60 year | 44.6 | 31.0 | 242.0 | 651.3 | 13.7 |
| Vitamin A RE [3] | % DRI 1–3 year | 1.4 | 1.43 | 6.4 | 0.00025 | 0.5 |
| | % DRI 19–60 year | 1.7 | 1.71 | 7.7 | 0.0003 | 0.6 |
| Vitamin E | % DRI 1–3 year | 1160.0 | 1070.0 | 0 | 970.0 | 25.0 |
| | % DRI 19–60 year | 1740.0 | 1605.0 | 0 | 1455.0 | 37.5 |

[1]: In the Congo Basin forest region, a child aged 1–3 years, his or her daily food intake is estimated to be 200 g, while a non- lactating non- pregnant active woman aged between 19 and 60 years, on average consumes 300 g daily [10,28]. [2]: Energy requirement for African population calculated as a function of estimated weight (60 kg) and height (1.55 m) among Africans [53]: Retinol equivalents (REs) (conversion factor 6:1 from β-carotene equivalents to RE) [51]. [3]: The nutrient contribution of an average portion of fresh fruit to nutrient intake recommendations of individuals was calculated and expressed as a percentage of the Dietary Recommended Intakes (DRI) for 1–3-year-old children and non-pregnant non-breastfeeding females [10,28].

## 5. Conclusions

The pulps of the three forest fruits of *G. lacourtiana*, *P. longifolia* and *A. lepidophyllus* and the mashed nuts of the *P. oleosa* and *P. oleosa* were nutritionally diverse, exhibiting high content of bioactive compounds including vitamins E and C. The fruits of *G. lacourtiana* had the highest bioactive contents of vitamins C, flavonoids and polyphenols, while *Panda oleosa* had the highest proanthocyanins and vitamin E. The nuts of *Panda oleosa* had the highest essential minerals content of iron, zinc, selenium, calcium and magnesium. Based on the nutritional values of these forest fruits, it can be concluded that the three forest fruits and the two nuts, can make considerable contributions towards meeting nutrient recommended intakes, for iron, zinc, magnesium, vitamins C and E. These forest fruits are also good sources of health promoting bioactive compounds. There is need to disseminate information to policy makers and development partners on the nutritional and bioactive compositions of these fruits and nuts to promote their consumption among the local populations. It is also recommended that the greater number of forest foods in Gabon and other African countries with unknown nutrient contents should be analysed and the results used for formulating policies governing food and nutrition security. There is need to integrate nutrition sensitive activities and indicators in forest conservation and management interventions of community government programs and development partner strategies, in order to enhance the nutrition of vulnerable communities such as women and children.

**Author Contributions:** All authors jointly conceived, designed and implemented the study. R.F. performed the analyses and wrote the first draft of this article, to which the rest of the other authors provided input.

**Funding:** This study was carried out within the framework of the "Beyond Timber project," under grant 5650155000601 from the Congo Basin Forest Fund of the African Development Bank to Bioversity International and by the CGIAR Research Programme on Forests, Trees and Agroforestry.

**Acknowledgments:** The authors are grateful to all participants in the surveys, including colleagues from IRET and the key informants in forest concessionaires of CEB and CPAET-Bayonne. Our most sincere gratitude goes to the forest concessionaires that allowed entering into their forest concessions, the villagers that patiently sacrificed the time in responding to lengthy questionnaires and the Ministry of forestry for their outstanding collaboration.

**Conflicts of Interest:** The authors declare no conflict of interest.

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
