# Peer review of "Nutrient and Bioactive Composition of Five Gabonese Forest Fruits and Their Potential Contribution to Dietary Reference Intakes of Children Aged 1–3 Years and Women Aged 19–60 Years"

_forests, doi:10.3390/f10020086_

Round 1

Reviewer 1 Report

The paper is interesting.

I have to main critical points:

1. please provide the full statistical analysis in the tables. You are showing the SD (standard deviation). But you don't inform whether the levels of the compounds in the studied fruits are significantly different between the species or not. And it is important before you start to conclude.

2. please explain why the local population in Gabon is not using enough the forest fruits. You conclude properly that the policy should be into increasing intake of the local forest fruits by the Gabon citizens. But you don't explain why they don't eat enough these fruits now, although these fruits are growing in the forests in the sufficient yields, as I understand.

I think that I want to see the paper after the improvement.

Author Response

Dear Danae, Below is our response to the reviewer’s' comments for the Manuscript "Nutrients and bioactive compounds content of “Nutrient and bioactive composition of five Gabonese forest fruits and their potential contribution to dietary reference intakes of children aged 1-3 years and women aged 19-60 years” Generally, in response to the comments raised by the reviewer, in the revised manuscript, we have used red font to indicate the adjustments, additions and reediting of the sentences. Reviewer One The paper is interesting. I have two main critical points: 1. Please provide the full statistical analysis in the tables. You are showing the SD (standard deviation). But you don't inform whether the levels of the compounds in the studied fruits are significantly different between the species or not. And it is important before you start to conclude. Full statistical analyses have been provided in the tables 1, 2 and 3, to show how significantly different the compounds among the species. In each table a footnote has been introduced to explain how the species differ and how the levels of significance were arrived at. Lines 248-249 under the methods section, data analysis sub section on page 13, have been introduced how the level of significance among the species compounds was calculated. Under the results, we have introduced lines 261-262 under proximate composition (paragraph 3.1), lines 269-273 under minerals (paragraph 3.2) and line 276 under bioactive compounds (paragraph 3.3), to show how nutrients are significantly vary. 2. Please explain why the local population in Gabon is not using enough the forest fruits. You conclude properly that the policy should be into increasing intake of the local forest fruits by the Gabon citizens. But you don't explain why they don't eat enough these fruits now, although these fruits are growing in the forests in the sufficient yields, as I understand. I think that I want to see the paper after the improvement. Lines 44-47 and lines 52-54 on page 3, of the introduction, have been provided to explain why the local Gabon population does not consume adequate forest foods

Reviewer 2 Report

Robert Fungo et al measured the nutrient and bioactive composition of five Gabonese forest fruits and discussed their potential contribution to dietary reference intakes of local population.  This is an interesting study and could have an impact to local community.

Minor:

RDI or DRI? The authors need to double check it through the manuscript.

Author Response

Dear Danae,

Below is our response to the reviewer’s' comments for the Manuscript "Nutrients and bioactive compounds content of “Nutrient and bioactive composition of five Gabonese forest fruits and their potential contribution to dietary reference intakes of children aged 1-3 years and women aged 19-60 years

Generally, in response to the comments raised by the reviewer, in the revised manuscript, we have used red font to indicate the adjustments, additions and reediting of the sentences.  

Reviewer Two

Comments and Suggestions for Authors

Robert Fungo et al measured the nutrient and bioactive composition of five Gabonese forest fruits and discussed their potential contribution to dietary reference intakes of local population.  This is an interesting study and could have an impact to local community.

Minor:

RDI or DRI? The authors need to double check it through the manuscript.

We have double checked and RDI has been replaced with DRI, throughout the manuscript. 

Reviewer 3 Report

The target population of dietary intake calculation should be specified also in the title.

Lines 41-42 should be discussed with major argumentations.

At lines the "obesity" should be described with related references.

At line 48 definition of sustainable consumption should be given.

At line 59  description of Gabone forest fruits should be inserted: plant , nutrient and bioactive components profile.

A graphic of sampling should be inserted.

Method of fat, fiber and protein should be better described and references added.

The authors should justify the choice of study specific bioactive compounds.

Statistical Analysis should be added for Table 1, 2, 3 and description of results in paragraph 3.1, 3.3 and 3.3 should be implemented.

Paragraph 4.3 should be divided in subparagraph for each class of compounds.

A paragraph on  data correlation should be added. 

Author Response

Dear Danae,

Below is our response to the reviewer’s' comments for the Manuscript "Nutrients and bioactive compounds content of “Nutrient and bioactive composition of five Gabonese forest fruits and their potential contribution to dietary reference intakes of children aged 1-3 years and women aged 19-60 years

Generally, in response to the comments raised by the reviewer, in the revised manuscript, we have used red font to indicate the adjustments, additions and reediting of the sentences

Reviewer Three

Comments and Suggestions for Authors

The target population of dietary intake calculation should be specified also in the title.

The target population of children aged 1-3 years and women aged 19-60 years has been included in the title.

Lines 41-42 should be discussed with major argumentations.

Lines, lines 44-47 on page 3 have introduced to improve on clarity of these lines.

Also reference 4 on page 26, has been introduced to address this comment.

At lines the "obesity" should be described with related references.

Lines 57-60 on page 3, describe obesity prevalence in Gabon and how it is a public health problem in Gabon. The related references are 5 and 8.

References 5 and 8 are on page 26

At line 48 definition of sustainable consumption should be given.

Lines 63-66 on page 3, have been introduced to provide the definition of sustainable consumption.

Also reference 9 on page 27, for the definition of sustainable consumption has been introduced.

At line 59 description of Gabone forest fruits should be inserted: plant, nutrient and bioactive components profile.

Lines 72-73 have been introduced to address this comment. Lines 73-80 also, further explain how the nutrient profile of forest foods can address nutrient deficiencies in Gabon and neighboring countries in the Congo Basin.

A graphic of sampling should be inserted.

Figure 3 on page 8, with a schematic representation of the sampling of fruits in the field and in the Laboratory, has been introduced.

Method of fat, fiber and protein should be better described and references added.

Detailed methods for fat, fiber and protein analysis have been added in lines, 173-177 for fat analysis, lines 179-183 crude fat for crude fat analysis and 183-190 protein analysis.

In accordance with this comment, reference AOAC 18 method 978.10 for crude fiber, reference 18, AOAC Kjeldahl 984.13 method for protein and reference AOAC 18 ether extraction method 920.39, have been introduced accordingly.

The authors should justify the choice of study specific bioactive compounds.

The justification for the choice of bioactive compounds is provided in lines 73-80 on page 4.

Also, lines 427-436 on page 22 justify choice of bioactive compounds

Statistical Analysis should be added for Table 1, 2, 3 and description of results in paragraph 3.1, 3.3 and 3.3 should be implemented.

Full statistical analyses have been provided in the tables 1, 2 and 3, to show how significantly different the compounds among the species. In each table a footnote has been introduced to explain how the species differ and how the levels of significance were arrived at.

Lines 248-249 under the methods section, data analysis sub section on page 13, have been introduced how the level of significance among the species compounds was calculated.

Under the results, we have introduced lines 261-262 under proximate composition (paragraph 3.1), lines 269-273 under minerals (paragraph 3.2) and line 276 under bioactive compounds (paragraph 3.3), to show how nutrients are significantly vary.

Paragraph 4.3 should be divided in subparagraph for each class of compounds.

In accordance with this comment paragraph 4.3 has been sub divided according to each class of compound

A paragraph on data correlation should be added.

Paragraph 4.4 provides for data correlation has been included

Regards

Robert Fungo (PhD)

Round 2

Reviewer 1 Report

Now the paper is much better after the improvement. I have now only one but important remark. You explain the Gabonese population has very limited access to the forests and this is a reason why the citizens are not eating these valuable fruits for the forest.

But in the conclusions there is nothing about it. You prove in your study that these fruits are so valuable and you recommend to include them into a regular diet, but you don’t conclude how to do it if the access is so limited! Please add the adequate sentence at the end of discussion and in the conclusions.

Author Response

Editor

Journal of Forestry

Dear Danae,

Below is our response to the reviewer’s' comments for the Manuscript "Nutrients and bioactive compounds content of “Nutrient and bioactive composition of five Gabonese forest fruits and their potential contribution to dietary reference intakes of children aged 1-3 years and women aged 19-60 years

Generally, in response to the comments raised by the reviewer, in the revised manuscript, we have used red font to indicate the adjustments, additions and reediting of the sentences.  

Reviewer One

Now the paper is much better after the improvement. I have now only one but important remark. You explain the Gabonese population has very limited access to the forests and this is a reason why the citizens are not eating these valuable fruits for the forest.

But in the conclusions there is nothing about it. You prove in your study that these fruits are so valuable and you recommend to include them into a regular diet, but you don’t conclude how to do it if the access is so limited! Please add the adequate sentence at the end of discussion and in the conclusions.

In accordance with the above comment, lines 36-37 in the abstract, lines 463-465 at end of the discussion section and lines 488-492 in the conclusion section have been introduced.

Reviewer Two

No comment.

Reviewer Three

No comment

Regards

Robert Fungo (PhD)

Reviewer 3 Report

The authors have improved the manuscript following the suggestions and it is now suitable for publication

Author Response

NO Comment
